# Role of Epstein-Barr Virus and Human Papillomavirus Coinfection in Cervical Cancer: Epidemiology, Mechanisms and Perspectives

**DOI:** 10.3390/pathogens9090685

**Published:** 2020-08-21

**Authors:** Rancés Blanco, Diego Carrillo-Beltrán, Julio C. Osorio, Gloria M Calaf, Francisco Aguayo

**Affiliations:** 1Programa de Virología, Instituto de Ciencias Biomédicas (ICBM), Faculty of Medicine, Universidad de Chile, Santiago 8380000, Chile; rancesblanco1976@gmail.com (R.B.); diegocb17@hotmail.com (D.C.-B.); 2Population Registry of Cali, Department of Pathology, Universidad del Valle, Cali 760042, Colombia; cejulio704@gmail.com; 3Instituto de Alta Investigación, Universidad de Tarapacá, Arica 1000000, Chile; gmc24@cumc.columbia.edu; 4Center for Radiological Research, Columbia University Medical Center, New York, NY 10032, USA; 5Universidad de Tarapacá, Arica 1000000, Chile; 6Advanced Center for Chronic Diseases (ACCDiS), Facultad de Medicina, Universidad de Chile, Santiago 8330024, Chile

**Keywords:** Epstein-Barr virus, human papillomavirus, coinfection, cervical cancer, tissue-infiltrating lymphocytes

## Abstract

High-risk human papillomavirus (HR-HPV) is etiologically associated with the development and progression of cervical cancer, although other factors are involved. Epstein-Barr virus (EBV) detection in premalignant and malignant tissues from uterine cervix has been widely reported; however, its contribution to cervical cancer development is still unclear. Here, a comprehensive analysis regarding EBV presence and its potential role in cervical cancer, the frequency of EBV/HR-HPV coinfection in uterine cervix and EBV infection in tissue-infiltrating lymphocytes were revised. Overall, reports suggest a potential link of EBV to the development of cervical carcinomas in two possible pathways: (1) Infecting epithelial cells, thus synergizing with HR-HPV (direct pathway), and/or (2) infecting tissue-infiltrating lymphocytes that could generate local immunosuppression (indirect pathway). In situ hybridization (ISH) and/or immunohistochemical methods are mandatory for discriminating the cell type infected by EBV. However, further studies are needed for a better understanding of the EBV/HR-HPV coinfection role in cervical carcinogenesis.

## 1. Introduction

Cervical cancer constitutes the fourth most frequently diagnosed malignant tumor in women and the first cause of cancer-related death in females. In 2018, 570,000 new cases and 311,000 deaths were estimated worldwide [1]. This is the leading diagnosed malignancy in 28 countries and the most frequent cause of cancer deaths in 42 countries, mainly in low- and middle-income countries [2]. More than 95% of cervical tumors arise from epithelial cells and include squamous cell carcinoma (SCC), adenocarcinoma and adenosquamous carcinoma.

The most important risk factor for cervical cancer development is infection with human papillomavirus (HPV) [3], with 99.7% of cervical carcinomas worldwide caused by high risk (HR)-HPV types, such as HPV16 or HPV18 [4,5]. While vaccines exist that protect against oncogenic HPV infection, global disparities still remain due to high costs [6,7]. However, HR-HPV infection is insufficient for cervical cancer development since low-grade squamous intraepithelial lesions (LSIL) usually regress to normal cells in cervical cytology or atypical squamous cells of undetermined significance (ASCUS). Only 3.6% of LSILs progress to high-grade squamous intraepithelial lesion (HSIL) [8], with a report of 50% regression in LSILs associated with HPV16 or HPV18 [9]. Accordingly, other host or environmental factors are necessary for cervical lesion progression, such as tobacco smoking (TS), oral contraceptive pills or immunosuppression, in particular infection with human immunodeficiency virus (HIV). Numerous studies have reported the simultaneous presence of Epstein-Barr virus (EBV) and HR-HPV in cervical carcinomas [10,11,12,13,14], although a potential role of EBV/HR-HPV coinfection in its development has not been established. Here, we review the current literature regarding EBV presence in SILs and cervical cancer together with its potential contribution to HR-HPV-mediated cervical carcinogenesis and tumor progression. In addition, we propose an HR-HPV/EBV co-carcinogenesis model, in which HR-HPV/EBV oncoproteins play a key role in both oncogenesis and immune evasion.

## 2. EBV Replication and Role in Cancer

EBV, also known as human herpesvirus 4 (HHV-4) is a gammaherpervirus that infects more than 90% of the human population worldwide [15]. It is characterized by a double-stranded DNA genome of 172 kb in length, surrounded by an envelope carrying various surface glycoproteins, with tegument proteins filling the space between the membrane and the inner icosahedral capsid [16]. EBV is related to some B cell-derived malignancies such as Burkitt’s lymphoma (BL) and Hodgkin’s disease (HD). Importantly, EBV is associated with some epithelial tumors, including undifferentiated nasopharyngeal carcinomas (NPCs) and a subset of gastric carcinomas (GCs) [15]. Once EBV virions achieve primary infection of B-cells, the virus establishes two phases of infection known as latent and lytic. In latency, the viral genome persists in the nucleus as an episome [17] with a small subset of ~90 coding regions expressed as: The EB-nuclear antigens (EBNAs) 1, 2, 3A, 3B, 3C and LP; the latent membrane proteins (LMPs) 1, 2A and 2B; the EB-encoded small RNAs (EBERs) 1 and 2 and ~50 different mature miRNAs [18,19]. These miRNAs cluster in two regions located on opposite sides of the EBV genome named the BamHI fragment H rightward open reading frame 1 (BHRF1) and BamHI A rightward transcripts (BART) regions. In particular, latency 0 occurs in non-dividing memory B-cells characterized by EBV genome persistence without viral protein expression with few non-coding RNAs (EBERs and BARTs) [20]. Latency I occurs in Burkitt lymphoma and GC, with only EBNA1, EBERs and BARTs being expressed. During latency IIa, EBNA1, the three LMPs, the two EBERs and BARTs are expressed, and in latency IIb, EBNA2 is also detected but LMPs are most restricted [21,22]. Latency II is mostly found in T/NK cells and classical HLs as well as in NPCs [23,24]. Latency III is characterized by expression of latent viral proteome including all EBNAs, all LMPs and EBERs, BHFR1s and BARTs, and witnessed in some lymphoproliferative disorders (reviewed in: [19,25]). Finally, the activation of the EBV lytic cycle produces viral progeny allowing the virus to spread from cell to cell and transmitting to new hosts [26]. Figure 1 shows the EBV genome organization with locations of latent genes.

The mechanisms of EBV-mediated B-cell cancers are well known (reviewed in [27]), although they are less understood in epithelial cells. In NPC cells, EBNA1 has a key role in EBV persistency, decreasing p53 accumulation in response to DNA damage [28], whilst inducing epithelial mesenchymal transition (EMT), deregulating some related genes [29] and enhancing angiogenesis in vitro [30]. LMP1, another latent protein, impairs the nuclear factor kappa-light-chain-enhancer of activated B cells (NF-κB) [31], activator protein 1 (AP-1) [32] and the Janus kinase/signal transducers and activators of transcription (JAK/STAT) [33] signaling pathways, leading to cell cycle disruption. Additionally, EBV-related epithelial tumors express BamHI-A rightward frame 1 (BARF1) [34], encoding a 220 amino acid lytic protein, secreted by EBV-infected epithelial cells as a soluble hexameric molecule (sBARF1) [35]. This viral protein hijacks human colony-stimulating factor 1 (hCSF-1), interfering with monocyte maturation [36] and impairing host immune responses against viral infections. It has been reported that BARF1 upregulates the expression of RelA and cyclin D1 and is able to reduce the cell cycle inhibitor p21WAF1 [37,38], promoting cell proliferation. Moreover, BARF1 activates the extracellular signal-regulated protein kinases 1 and 2 (ERK1/2)/c-Jun pathway [39] inhibiting apoptosis whilst increasing Bcl-2 and reducing both caspases and Bax [40]. Additionally, it inhibits interferon-alpha (IFN-α) production and release by mononuclear cells [41], impairing host immune responses. Interestingly, telomerase activation is observed in HPV-positive cervical cancer cells (HeLa) transfected with the BARF1 gene [42]. As BARF1 is expressed in EBV-associated NPC and GCs and not in lymphomas, BARF1 is considered an exclusive epithelial oncoprotein [42,43].

## 3. HPV in Cervical Cancer

HPV is a non-enveloped and exclusively intraepithelial virus, which comprises an 8 kb double-stranded circular DNA containing eight protein-coding genes divided into three major regions: Early, late and a noncoding region, known as the long control region (LCR). Early gene encoding non-structural proteins (E1 to E7) are involved in viral replication, transcription and transformation, while late transcribed genes encode for structural L1 and L2 proteins (reviewed in [44]). More than 210 HPV types have been identified by L1 sequencing, classified in HR-HPV (e.g., 16, 18 and 31) and low-risk (LR)-HPV types (e.g., 6, 7 and 11) according to the oncogenic potential [45,46]. Figure 1 shows the HPV genome organization.

HR-HPV infection is etiologically associated with cervical carcinomas, anogenital and a subset of head and neck squamous cell carcinomas (HNSCCs). Integration of HR-HPV genomes into the host is an important hit, although not a requisite for epithelial carcinogenesis [47,48]. This is common in HSIL and cervical SCC compared with normal or LSIL tissues [49]. Moreover, HR-HPV (e.g., 16 and 18) integration in HSIL is frequently accompanied by chromosomal abnormalities [50], with E2 region loss during integration, leading to constitutive E6 and E7 protein expression [51]. E6 is a ~150 amino acid protein comprising two zinc finger binding domains connected by a 36 amino acid long-linker, and the carboxy terminal domain containing a PDZ-binding motif interacting with cell proteins [52]. E6 promotes loss of p53 via E6-associated protein (E6-AP)-mediated ubiquitination and proteasome degradation, inhibits apoptosis [53] and activates Mitogen-activated protein kinase (MAPK) and the mechanistic target of rapamycin complex 1 (mTORC1) pathways [54,55]. Additionally, E6 targets the pro-apoptotic proteins Bak for degradation [56] and others such as the Fas-associated protein with death domain (FADD) and caspase-8, disrupting the apoptotic program [57]. Likewise, E7 is a 100 amino acid protein comprising three conserved regions denoted CR1, CR2 and CR3, with CR3 containing two CXXC motifs separated by 29 or 30 residues. Moreover, the E7 carboxyl terminal domain contains a zinc-binding motif mediating interaction with cellular proteins [58]. E7 targets retinoblastoma protein (pRb) for ubiquitination, leading to E2F transcription factor release, allowing cell entry to S-phase [59]. Both E6 and E7 interact with c-Myc inducing human telomerase reverse transcriptase (*hTERT*) promoter activation and *hTERT* expression, enabling cell evasion of senescence [60,61]. E6 and E7 also upregulate expression of EMT markers such as N-cadherin, Fibronectin and Vimentin, increasing cell migration and invasiveness [62]. Moreover, E6 and E7 inhibit interferon (IFN) antiviral activity and decrease tumor necrosis factor alpha (TNF-α) and (interleukin-1 beta) IL-1β secretion by macrophages, enabling immune evasion [63].

HR-HPV infection is insufficient for cervical carcinogenesis and involvement of other cofactors is essential. While factors involved in HR-HPV integration into the host genome are unclear [64], a prerequisite is DNA damage [65], TS being a very important factor for cervical cancer development [66] by activating the HPV16 p97 promoter and leading to E6 and E7 overexpression [67]. Moreover, HPV16 E6 and E7 collaborate with TS, increasing the tumor properties of epithelial cells [68,69], although other potential cofactors such as chronic inflammation and bacterial or viral coinfection are also suspected.

## 4. Frequency of HPV and EBV Coinfection in Uterine Cervix

To discriminate the linage of EBV infected cells (epithelial and/or lymphocytes), in situ hybridization (ISH) and immunohistochemistry (IHC) are important detection methods. Thus, EBV was detected in less than 16.7% of normal and non-tumor cervix samples [70,71,72]. In contrast, some studies show increased EBV frequency in cervical cancer ranging from 27.8% to 100% [70,71,73]. EBV infection appears in 62.5% and 27.8% of cervical carcinomas using BamHI O/K fragments, or BamHI W region ISH [70,71], respectively, whereas the EBV infection rate is 50.0% and 85.7% of cervical cancer by using EBER1 and BamHI W ISH, respectively [72]. Subsequently, 100% and 87.5% recorded positive in cervical cancer using the BamHI W region and EBNA2 ISH, respectively, confirming EBNA2 expression by immunofluorescence at 68.7% [73]. Additionally, EBNA2 and LMP1 expression (EBV latent-cycle proteins) is observed in 88.9% and 66.7% of cervical cancer [72], together with EBNA1 and LMP1 also detected by IHC [13].

When LSIL and HSIL are compared, EBV positive samples increase. EBV infection was found in 0%, 8.0% and 8.0% of cervical intraepithelial neoplasms (CIN) I, II and III, respectively [71], while another study revealed EBV presence in 33.3% and 70.0% of CIN I-II and CIN III patients, respectively [74]. In addition, EBV infection was detected in 20.0% and 41.7% of CIN II and CIN III, respectively, using EBER1 ISH, although such increase was not evidenced when ISH of the BamHI W region was analyzed [72]. These data suggest a potential contribution of EBV as a cofactor to the development of cervical carcinomas. Nevertheless, other studies did not detect EBV infection in premalignant or malignant epithelial cells [14,75,76,77,78,79], opposite results possibly related to differences in the tissue type, lesion extension or sensitivity of analytical assays. Accordingly, less than one copy of the EBV genome in cervical cancer samples was reported with a decreased number of EBV-positive cells observed by ISH and IHC, suggesting that only a few malignant epithelial cells are infected with this virus [72].

EBV/HR-HPV coinfection in SILs and cervical carcinomas ranges from 12.7% to 81.8% [10,11], whereas EBV was frequently associated with HPV16 and HPV18 [80] and increased the risk of HPV16 integration into the host genome [11,12]. Moreover, HPV+/EBV+ cervical cancer displays increased *RB1* and E-cadherin (*CDH1*) gene promoter methylation compared with HPV+/EBV− tumors [81]. However, a study conducted by Lattario et al. failed to find a relation between EBV and HPV coinfection and the methylation of death-associated protein kinase (*DAPK*) gene promoter in HSIL [82]. In HIV and HR-HPV infected patients, the additional infection with EBV significantly increased the risk of SIL compared to uninfected women [83,84]. Furthermore, there was no association of EBV DNA with a decreased count of CD4+ T lymphocytes with high HIV viral load [84,85], suggesting that EBV infection in cervical epithelium is independent of HIV status. In this respect, there was no difference when the frequency of EBV positivity in HIV-negative and HIV-positive patients was compared [86]. Remarkably, a common limitation for most of the aforementioned studies is the use of PCR for EBV detection [10,12,81,82,85], masking the biological significance of EBV/HR-HPV coinfection in epithelial cells. Contamination of collected cervical tumors with EBV-positive infiltrating lymphocytes may contribute to the EBV DNA detection reported in these tissues. In fact, 44.4% (8/18) of EBV infected cervical SCCs by PCR was found and only 27.8% (5/18) when ISH was used [71]. Notably, 21.7% (13/60) of EBV infected cervical SCCs by PCR was reported, but ISH only revealed positive signal in infiltrating lymphocytes [76]. Similar results were obtained by Seo et al. [79], although one report conducted by Abudoukadeer et al. demonstrated a high coincidence degree between PCR and IHC for LMP1 (k = 0.799) in their series of patients [87].

To the best of our knowledge, reports demonstrating unequivocal EBV/HR-HPV coinfection in cervical epithelial cells are scarce. For instance, an increased EBV/HR-HPV coinfection in HSIL and cervical SCC when compared with normal tissues and LSIL was detected [13]. Moreover, EBV/HR-HPV coinfection in 34.1% (15/44) of cervical SCC was found [88]. In this study, we demonstrated the relation of LMP1 (EBV) and E6 (HR-HPV) with poorly differentiated and invasive cervical SCCs as well as with the overexpression of the inhibitor of DNA binding 1 (Id-1) protein [88], which is associated with tumor formation and progression in different human malignancies, including cervical cancer [89,90]. In a similar way, there was no EBV/HR-HPV coinfection in normal and non-tumor tissues (0/12) while the simultaneous presence of these viruses was evidenced in 30.8% (4/13) and 69.2% (9/13) of cervical carcinomas when EBV was detected by ISH for EBER1 and BamHI W, respectively. EBV/HR-HPV coinfection was also found in 18.2% (2/11) (EBER1) and 27.3% (3/11) (BamHI W) of CIN II-III [72]. Furthermore, EBV/HR-HPV coinfection in 31.8% (7/22) of CIN I-II and CIN III was demonstrated, which was statistically significant when compared with non-premalignant tissues [74]. Similarly, LMP1 and HPV L1 proteins were detected in 30.0% (3/10) and 66.7% (4/6) of CIN I and CIN II, respectively, in HIV-positive patients [91]. Taken together, these results suggest that EBV/HPV coinfection is associated with epithelial cocarcinogenesis as well as with the development and progression of cervical cancer. Studies reporting EBV presence or EBV/HPV coinfections in cervical cancer are shown in Table 1.

## 5. EBV Infection in Tumor-Infiltrating Lymphocytes from Cervical Carcinomas

EBV infection in tumor-infiltrating lymphocytes was also reported in cervical SCCs [71,72,88], while no EBV-infected lymphocytes were observed in normal cervical samples [71,72] or uterine fibroids [79]. While EBV infection was detected in lymphocytes from non-tumor cervical tissues (3 chronic cervicitis, 2 cervical polypus and 2 hyperkeratosis) using ISH for EBER1 (57.1%) and BamHI W (71.4%), there was no HPV infection evidenced in these samples [72]. Notably, there was an absence of EBV infection in neoplastic cells, while this virus was only detected in infiltrating lymphocytes from both SILs and cervical SCC using ISH for EBER [76].

Interestingly, an increased number of EBV-positive infiltrating lymphocytes in CIN III (15.7%) and cervical SCC (15.0%) compared with CIN I/II (6.8%) was reported. Furthermore, the presence of EBV DNA detected by PCR was positively related with the degree of lymphocyte infiltration [76]. In the same way, an increased prevalence of EBV infection in infiltrating lymphocytes from patients with HSIL and cervical SCC was found when compared with no squamous intraepithelial lesion and LSIL [14]. These findings suggest that progression from LSIL to HSIL and cervical SCC is accompanied by an increased number of EBV-infected surrounding lymphocytes. However, an absence of EBER positive lymphocytes infiltrating the cervical stroma in CIN III and cervical SCC samples has also been reported [79].

EBV modulates the host immune response by diverse mechanisms [36,92], in part through the lytic cycle protein BCRF1 (viral IL-10), inhibiting production of some molecules by CD4+ T lymphocytes such as IFN-γ, IL-2 and IL-6 [93]. Remarkably, EBV shedding increases along with cervical SCC progression. Moreover, EBV-specific killer T cell activity decreases in advanced stages of cervical cancer compared with earlier stages, demonstrating an impaired T cell immunity [94]. In addition, HR-HPV E6 and E7 inhibit IFN antiviral activity and decrease secretion of TNF-α and IL-1β by macrophages [63]. Accordingly, immunosupression generated by EBV infection could contribute to immune evasion of HPV-infected epithelial cells, although further studies are required (Figure 2).

## 6. Mechanisms of HPV/EBV-Mediated Cervical Carcinogenesis

Experimental approaches evaluating molecular mechanisms involved in EBV/HR-HPV coinfection are limited. However, EBV LMP1 in combination with HPV16 E6 viral proteins in transformed mouse embryonic fibroblasts (MEFs) reduces components of DNA damage response (DDR) such as p53, pRb and p27, whilst increasing checkpoint kinase 1 (CHK1), NF-κB signaling, v-akt murine thymoma viral oncogene (Akt) and MAPK signaling [95,96]. In addition, LMP1 induces down-regulation of E-cadherin expression [97] and also regulates TWIST [98] and SNAIL [99] transcription factors and others related with cell motility. Moreover, LMP1 and E6 co-expression induces cell proliferation, resistance to apoptosis, anchorage-independent growth and tumor-formation ability in nude mice compared with single expression of EBNA1 or E6 [95]. In EBV-infected NPC cells, EBNA1 plays a role in EMT through inhibition of both microRNA (miR)-200a and -200b expression, while it up-regulates the expression of *ZEB1* and *ZEB2*, their target genes [29]. Interestingly, double expression of LMP1 and HR-HPV E6 relates with a more aggressive form of malignant tumor such as breast adenocarcinoma [100] and cervical SCC [88]. Thereby, in the case of EBV-associated GCs, EBNA1 may induce constitutive and also IFNγ-inducible programmed death-ligand 1 (PD-L1) expression in EBV-infected epithelial-cells [101]. In HPV+ HeLa and SiHa cervical cells, transfection with a small interfering RNA (siRNA) for Myc knock down was able to reduce the *hTERT* promoter activity [102]. HPV16 E6 activates *hTERT* gene transcription through induction of c-Myc, which is overexpressed in cervical carcinomas [103,104]. Moreover, *hTERT* activation in BARF1-transfected PATAS monkey kidney cells was accompanied by up-regulation of c-Myc, while in HeLa cells BARF1 induces activation of telomerase binding directly to initiator elements in the *hTERT* promoter region [42]. Alternatively, both HPV18 E6 and E7 oncoproteins are necessary to increase EBV genome maintenance in normal oral keratinocytes (NOKs) and to induce the reactivation of the EBV lytic program in suprabasal layers of oral epithelial from an in vitro organotypic raft culture model [105]. However, a significant reduction in EBV immediate early (BZLF1, BRLF1) and early (BALF5, BMRF1) gene expression with increased expression of EBER1 was evidenced in HPV+/EBV+ human foreskin keratinocytes (HFK) compared to that in HPV−/EBV+ rafts [106]. In cervical cells, increased levels of EBER1 may contribute to the transition from inflammation to oncogenesis of HPV-associated cervical cancer by modulating innate immune responses [107]. In addition, LMP1 reduces the presence of HPV16 with no expression change in the EBNA1 and EBNA2 latent genes, suggesting that EBV latency is favored over lytic replication in HPV16+ cells. However, LMP1 is mostly expressed in B-cell lymphomas, lymphoproliferative disorders and NPC [25], but less in cervical and GCs [72,108]. In uterine cervix, latency III EBV infection is increased in CIN and cervical SCC compared with non-malignant samples [72], conferring a long-term persistence of EBV infection in malignant cells. Furthermore, reduced EBV replication mediated by HPV16 E7 is related to retarded expression of some markers related with early epithelial cell differentiation [106]. Similarly, cervical SCC EBV/HR-HPV coinfection is associated with decreased cell differentiation [88], closely related with a more aggressive tumor. Additionally, NOKs and FaDu hypopharyngeal carcinoma cells coinfected with EBV and HR-HPV show increased invasiveness in the lysophosphatidic acid (LPA) presence (glycerophospholipid able to stimulate cell migration) compared to EBV−/HR-HPV- and EBV+/HR-HPV- cells [109], resembling the effects of coinfection in cervical SCC [88]. According to our knowledge, the levels of BARF1 in cervical cancer have been examined in only one study, which reported 27% detection in tumors from Algerian women [13]. As previously stated, this protein has been suggested as an epithelial EBV oncogene, which is expressed in a cell differentiation-dependent manner [110]. Due to this lytic protein being expressed in a majority of EBV+ NPC and GC cells, the possibility that this protein is involved in EBV+ cervical carcinomas, working in both oncogenic process and immune evasion is plausible. Considering that EBV infection in normal epithelial cells is exclusively lytic [111,112], an interesting point is the establishment of EBV latency in epithelial cervical cells. In this respect, it has been suggested that previous DNA damage in cells is a requisite [113]. Thus, we can propose a scenario in which previous HPV infected and subsequently DNA-damaged cells (e.g., by HPV E7) are particularly susceptible to EBV latency establishment. Moreover, we can speculate on the possibility that HPV is involved in the activation of the EBV abortive lytic cycle, promoting expression of some lytic genes, such as Zebra or Rta, which has been reported to be important in EBV-mediated carcinogenesis. A hypothetical model in which HPV infection favors EBV latency establishment in epithelial cervical cancer cells is shown in Figure 3.

## 7. Conclusions and Remarks

EBV/HR-HPV coinfection was recently related with an increased risk of cervical cancer [114]. However, this analysis failed to discriminate the cell linage infected by EBV in cervical tissues (epithelial and/or lymphocytes). Overall, studies suggest a potential cooperation of EBV with cervical cancer development by two different mechanisms: (1) Infecting epithelial cells, possibly synergizing with HR-HPV (direct mechanism) and (2) infecting tissue-infiltrating lymphocytes and generating local immunosuppression (indirect mechanism). Further studies are needed for a better understanding of the role of EBV and HPV coinfection in the carcinogenesis of cervical epithelium, in which ISH and/or IHC are mandatory to determining the cell linage infected by EBV. Moreover, the potential role of circulating EBV DNA load in cervical cancer needs to be evaluated. Finally, in vitro and in vivo assays using cervical cell lines coexpressing EBV and HPV oncoproteins should be conducted to confirm whether coinfection with these viruses synergizes to the development of cervical cancer. More studies are warranted to evaluate some EBV lytic genes such as BARF1 in cervical carcinomas and the potential role of EBV abortive lytic infection in HPV-mediated cervical carcinogenesis.

## Figures and Tables

**Figure 1 pathogens-09-00685-f001:**
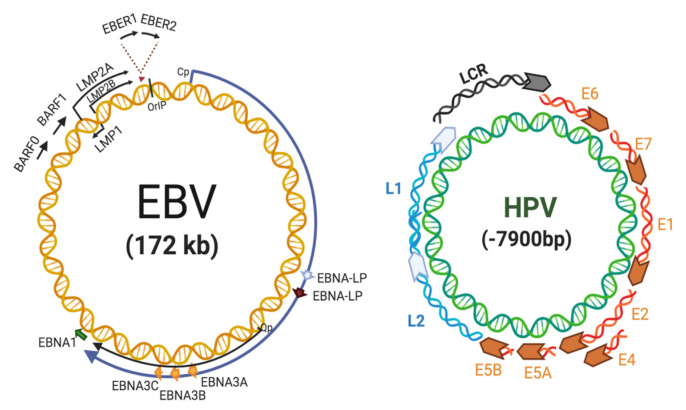
Epstein-Barr virus (**left**) and human papillomavirus (**right**) genome organizations. Created by BioRender.com.

**Figure 2 pathogens-09-00685-f002:**
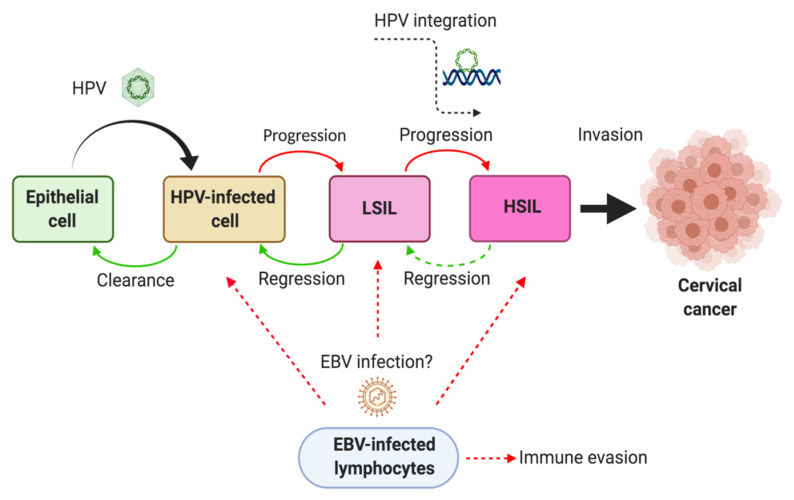
A model of carcinogenesis induced by high risk (HR)-HPV/EBV coinfection in the uterine cervix. HR-HPV infected cells are prone to latent EBV infection during neoplastic progression. Additionally, EBV infection of infiltrating lymphocytes cooperates with the immune escape of HPV-infected cells, modulating the host immune responses. Created by BioRender.com.

**Figure 3 pathogens-09-00685-f003:**
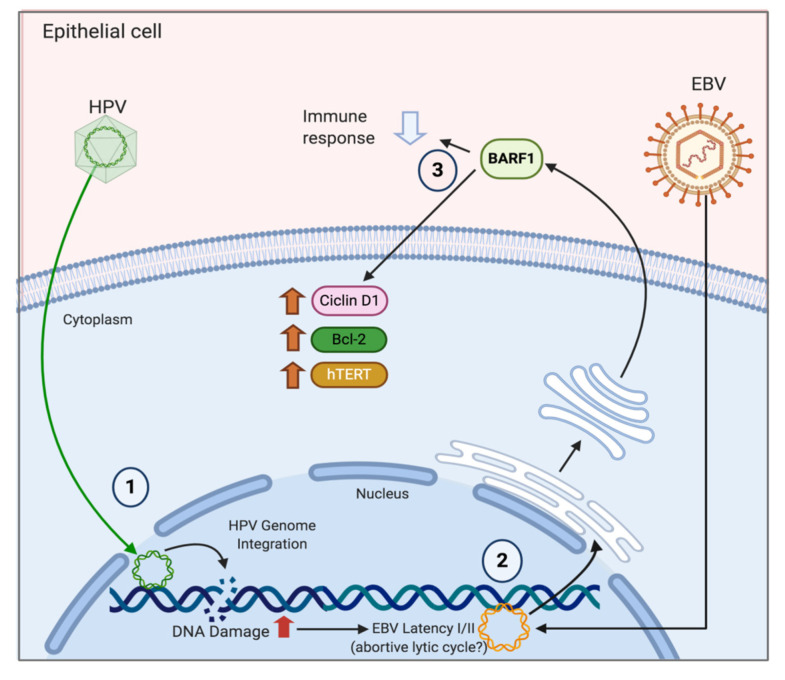
A hypothetical model of HR-HPV/EBV interaction in cervical epithelial cells. **1**. HR-HPV infection and viral integration promotes DNA damage and genomic instability in cervical cells; **2**. DNA damage promotes the establishment of EBV latency (I/II) with potential expression of some lytic genes such as BARF1 (abortive lytic expression); **3**. BARF1 (and other viral oncogenes) promotes oncogenic changes and immune evasion, cooperating with HR-HPV. Created by BioRender.com.

**Table 1 pathogens-09-00685-t001:** Epstein-Barr virus (EBV) and human papillomavirus (HPV) coinfection detected in squamous intraepithelial lesions (SILs) and cervical carcinomas.

Ref.	EBV		HPV		EBV/HPV Coinfection
	Methods	Results	Methods	Results	
[70]	ISH of BamHI O/K	Normal cervix = 0/15 (0%)	-	-	-
		CIN I = 1/1 (100%)			
		Cervical cancer = 5/8 (62.5%)			
[71]	ISH of BamHI W	Normal cervix = 0/25 (0%)	-	-	-
		CIN I = 0/25 (0%)			
		CIN II = 2/25 (8.0%)			
		CIN III = 2/25 (8.0%)			
		SCC = 5/18 (27.8%)			
[72]	ISH of EBER1	Normal cervix = 0/5 (0%)	PCR for E6/E7	Normal cervix = 0/5 (0%)	Normal cervix = 0/5 (0%)
		CIN II = 1/5 (20.0%)		CIN II = 1/3 (33.3%)	CIN II = 0/3 (0%)
		CIN III = 5/12 (41.7%)		CIN III = 6/8 (75.0%)	CIN III = 2/8 (25.0%)
		Cervical cancer = 7/14 (50.0%)		Cervical cancer = 10/13 (76.9%)	Cervical cancer = 4/13 (30.8%)
	ISH of BamHI W	Normal cervix = 0/5 (%)			Normal cervix = 0/5 (0%)
		CIN II = 4/5 (80.0%)			CIN II = 0/3 (0%)
		CIN III = 8/12 (66.7%)			CIN III = 3/8 (37.5%)
		Cervical cancer = 12/14 (85.7%)			Cervical cancer = 9/13 (69.2%)
	IFI for EBNA2	Normal cervix = 0/3 (0%)			Normal cervix = 0/3 (0%)
		CIN III = 6/8 (75.0%)			CIN III = 3/6 (50.0%)
		Cervical cancer = 8/9 (88.9%)			Cervical cancer = 5/8 (62.5%)
	IFI for LMP1	Normal cervix = 0/3 (0%)			Normal cervix = 0/3 (0%)
		CIN III = 4/8 (50.0%)			CIN III = 2/6 (33.3%)
		Cervical cancer = 6/9 (66.7%)			Cervical cancer = 4/8 (50.0%)
[73]	ISH of BamHI W	Normal cervix = 0/2 (0%)	-	-	-
		CIN I = 2/2 (100%)			
		CIN II-III = 2/2 (100%)			
		Cervical cancer = 10/10 (100%)			
	ISH of EBNA2	Normal cervix = 0/3 (0%)			
		CIN I = 2/2 (100%)			
		CIN II-III = 2/3 (66.7%)			
		Cervical cancer = 14/16 (87.5%)			
	IFI for EBNA2	Normal cervix = 0/3 (0%)			
		CIN I = 0/2 (0%)			
		CIN II-III = 1/3 (33.3%)			
		Cervical cancer = 11/16 (68.7%)			
[74]	ISH of EBERs	CIN I-II = 4/12 (33.3%)	PCR-ELISA for MY09/MY11	CIN-negative = 2/26 (7.7%)	CIN I-II = 3/12 (25.0%)
		CIN III = 7/10 (70.0%)		CIN I-II = 5/12 (41.7%)	CIN III = 4/10 (40.0%)
				CIN III = 7/10 (70.0%)	
[91]	IHC for LMP1	CIN I = 1/10 (10.0%)	IHC for HPV	CIN I = 3/10 (30.0%)	CIN I = 3/10 (30.0%)
		CIN III = 3/3 (100%)		CIN II = 2/6 (33.3%)	CIN II = 4/6 (66.7%)
[13]	IHC for EBNA1	SCC = 8/23 (34.8%)	PCR/Hybrid Capture 2 (HC2)	Normal cervix = 2/14 (14.3%)	-
	IHC for LMP1	SCC = 6/23 (26.1%)		CIN I = 12/16 (75.0%)	
				CIN II-III = 20/21 (95.2%)	
				SCC = 51/58 (87.9%)	
[88]	IHC for LMP1	SCC = 15/44 (34.1%)	PCR for E6/E7	SCC = 42/44 (95.5%)	SCC = 15/44 (34.1%)

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
