# Peer review of "Role of Epstein-Barr Virus and Human Papillomavirus Coinfection in Cervical Cancer: Epidemiology, Mechanisms and Perspectives"

_pathogens, 2020, doi:10.3390/pathogens9090685_

Round 1

Reviewer 1 Report

The authors review the most recent research regarding cervical cancer, HPV, and EBV, which are generally under-studied topics. However, to bring the review up to publication standard, the following three recommendations are made.

  1. Construction of a review requires not only introducing literature that is novel relative to existing reviews, but the presence of a unique perspective. For example, Vranic et al. recently published an excellent review covering a very similar set of topics to the current review (Vranic S et al., Front. Oncol., 2018). It was difficult to identify the unique perspective provided by the current review, relative to past reviews.
  2. Furthermore, a high-quality review should include visually pleasing and easy-to-understand figures and tables. The major shortcoming of the current review is the inclusion of small images of low quality that are quite similar to those in existing reports. For example, Figure 1 is very similar to that in a review by Young and Rickinson (Young LS and Rickinson AB, Nat. Rev. Cancer, 2004). It is necessary to improve the figure’s quality and originality.
  3. Finally, the current review requires extensive editing for English language and grammar. For example, abbreviations should be defined at their first appearance in the text. In this case, Epstein-Barr virus (EBV) should be defined only once, followed thereafter by consistent use of the abbreviation (lines 48 and 54). Similarly, TNF (tumor necrosis factor), IL-1 (interleukin-1), and HNSCC (head and neck squamous cell carcinoma) should be defined at their first use in the text, followed thereafter by consistent use of the abbreviations. In addition, commas are missing from many sentences. For example, when listing multiple nouns using “and”, a comma should precede “and”. Therefore, it is strongly advised that the manuscript be carefully revised by an English language editor.

Author Response

Reviewer 1. The authors review the most recent research regarding cervical cancer, HPV, and EBV, which are generally under-studied topics. However, to bring the review up to publication standard, the following three recommendations are made.

Construction of a review requires not only introducing literature that is novel relative to existing reviews, but the presence of a unique perspective. For example, Vranic et al. recently published an excellent review covering a very similar set of topics to the current review (Vranic S et al., Front. Oncol., 2018). It was difficult to identify the unique perspective provided by the current review, relative to past reviews.

Answer. The authors appreciate the reviewer comment. We added sentences about mechanisms involved in the interaction between HR-HPV and EBV. In addition, we proposed potential mechanisms of HR-HPV/EBV-mediated oncogenesis, in which the viral protein BARF1 has an important role, as suggested in both EBV-associated gastric and nasopharyngeal carcinomas. As a perspective, we proposed the analysis of BARF1 in cervical carcinomas and the importance of EBV abortive lytic cycle, which could be modulated by HPV.

Reviewer 1. Furthermore, a high-quality review should include visually pleasing and easy-to-understand figures and tables. The major shortcoming of the current review is the inclusion of small images of low quality that are quite similar to those in existing reports. For example, Figure 1 is very similar to that in a review by Young and Rickinson (Young LS and Rickinson AB, Nat. Rev. Cancer, 2004). It is necessary to improve the figure’s quality and originality.

Answer. Many thanks for these observations. We included a new figure with proposed mechanisms in HR-HPV/EBV cervical carcinogenesis (Figure 3). In order to compare HPV and EBV genome organizations, the Figures 1 and 2 were combined; however, if the reviewers consider that these figures are not relevant, we can delete them. In addition, The Figure 2 was optimized.

Reviewer 1. Finally, the current review requires extensive editing for English language and grammar. For example, abbreviations should be defined at their first appearance in the text. In this case, Epstein-Barr virus (EBV) should be defined only once, followed thereafter by consistent use of the abbreviation (lines 48 and 54). Similarly, TNF (tumor necrosis factor), IL-1 (interleukin-1), and HNSCC (head and neck squamous cell carcinoma) should be defined at their first use in the text, followed thereafter by consistent use of the abbreviations. In addition, commas are missing from many sentences. For example, when listing multiple nouns using “and”, a comma should precede “and”. Therefore, it is strongly advised that the manuscript be carefully revised by an English language editor.

Answer. The manuscript was extensively reviewed for grammar mistakes, abbreviation use, etc. In addition, some expressions were corrected.

Reviewer 2 Report

The manuscript by “Rances Blanco et al” describes the epidemiology, molecular mechanisms of EBV and HPV coninfection in cervical cancer.

Specific comments:

Line 203-209.  There are inappropriate and old references (71,72,76, et al) to describle the previous work about " EBV infection in tumor infiltrating lymphocytes". Nowday, genomic tools (single-cell seqeuncing) are more popular to devect the molecular pathology of SILs and SCC.

Line 229-261. The activation of c-myc are essential for both EBV and HPV tumorigenesis. The findings will offer an explanation for the carcinogenic mechanism of these virus.

Author Response

Reviewer 2. The manuscript by “Rances Blanco et al” describes the epidemiology, molecular mechanisms of EBV and HPV coninfection in cervical cancer.

Specific comments:

Line 203-209.  There are inappropriate and old references (71,72,76, et al) to describle the previous work about " EBV infection in tumor infiltrating lymphocytes". Nowday, genomic tools (single-cell seqeuncing) are more popular to devect the molecular pathology of SILs and SCC.

Answer. The authors appreciate the reviewer comments. However, we consider that the use of references 71, 72, and 76 is necessary because in these studies the authors used in situ hybridization (ISH) which permit to discriminate the cell type infected by EBV (epithelial cells and/or lymphocytes). Up to now, ISH for EBERs remains as the gold standard for EBV detection in tissue samples. On the other hand, we agree with the reviewer, single-cell sequencing could be a valuable method to evaluate EBV infection in cervical samples. However, in the reviewed literature, we did not find any manuscript in which single-cell sequencing or other genomic tools were used for this purpose.

Reviewer 2. Line 229-261. The activation of c-myc are essential for both EBV and HPV tumorigenesis. The findings will offer an explanation for the carcinogenic mechanism of these virus.

Answer. The authors agree with the reviewer and appreciate this comment. In this sense, information concerning the role of c-myc activation in EBV and HPV infection was included in the manuscript.

Round 2

Reviewer 1 Report

  In the previous review, it was pointed out three drawbacks to this review. 1) originality, 2) quality of figures, and 3) quality of English. In this revise, a big improvement is recognized about the English sentence and originality. On the other hand, compared to last peer review, the improvement in the figure was almost not observed. It is strongly recommended that the editorial office also support specific improvements to the figure.

  Although the authors have added Fig. 3, all of figures and table also have the same problem, and the authors could not understand the fundamental problem of the figure.

  1. Figure and font size

  The text used in all figures and table can be understood by printing on paper. At the very least, it couldn't read the letters in Fig. 1, Table 1, and Fig. 3 from the printouts. These figures need to be about 1.2 to 1.5 times larger. The arrow in Fig. 2 is also very thin.

  1. Unified notation

  On the left of Fig. 1, the arrow is outside the viral genome, while on the right it is on or inside the genome. Also, the fonts in Figs. 1 and 2 appear to use different fonts. Moreover, in EBV, it is described as a double-stranded DNA episome, whereas in HPV, it is described as a papilloma virus. All diagrams must have the same pattern. In addition, Young has published a very similar diagram to the left of Fig.1. As with Fig/ 2, the origin of this figure should be specified.

  1. Inaccurate information

  For example, in the EBV diagram, the description of LMP2B is missing despite the arrow. Also, in Fig. 2, the EBV at the bottom of Fig. 2 is unnecessary. Because EBV infection is caused by direct contact between infiltrating EBV-positive lymphocytes and epithelial cells. Similarly, BARF1 is not expressed in latency I / II. BARF1 is expressed in some cells in which lytic infection is reactivated. Such incorrect statements are everywhere.

  To improve Table 1, the font size must be increased. There are many unnecessary entries in this table. For example, although the reference is described in Table 1, the space can be omitted by writing the reference number. Conclusion in the table should be explained in the text if it is really necessary. Also, part of the table has disappeared. Tissue can be Frozen or FFPE only. By making such an arrangement, it is possible to eliminate unnecessary information and increase the size of the font.

Author Response

REVIEWER 1. In the previous review, it was pointed out three drawbacks to this review. 1) originality, 2) quality of figures, and 3) quality of English. In this revise, a big improvement is recognized about the English sentence and originality. On the other hand, compared to last peer review, the improvement in the figure was almost not observed. It is strongly recommended that the editorial office also support specific improvements to the figure. Although the authors have added Fig. 3, all of figures and table also have the same problem, and the authors could not understand the fundamental problem of the figure.

ANSWER. We agree with these commentaries. The Figures were newly designed, and the Table was corrected.

REVIEWER 1. Figure and font size. The text used in all figures and table can be understood by printing on paper. At the very least, it couldn't read the letters in Fig. 1, Table 1, and Fig. 3 from the printouts. These figures need to be about 1.2 to 1.5 times larger. The arrow in Fig. 2 is also very thin.

ANSWER. Many thanks for these observations. The font size used in Fig. 1-3 and Table 1 was augmented. In addition, the size of Fig. 1-3 was figure size enlarged.

REVIEWER 1. Unified notation. On the left of Fig. 1, the arrow is outside the viral genome, while on the right it is on or inside the genome. Also, the fonts in Figs. 1 and 2 appear to use different fonts. Moreover, in EBV, it is described as a double-stranded DNA episome, whereas in HPV, it is described as a papilloma virus. All diagrams must have the same pattern. In addition, Young has published a very similar diagram to the left of Fig.1. As with Fig/ 2, the origin of this figure should be specified.

ANSWER. Many thanks for these observations. Fig. 1 (left and right) was completely changed. In the new Figure, the same font was used. Also, to homogenize Fig. 1, the viruses were described as EBV and HPV.

REVIEWER 1. Inaccurate information. For example, in the EBV diagram, the description of LMP2B is missing despite the arrow. Also, in Fig. 2, the EBV at the bottom of Fig. 2 is unnecessary. Because EBV infection is caused by direct contact between infiltrating EBV-positive lymphocytes and epithelial cells.

ANSWER. Many thanks for these observations. In the newer Fig. 1 (left) LMP2B was included. The EBV at the bottom of Fig. 2 was removed and the figure was changed.

REVIEWER 1. Similarly, BARF1 is not expressed in latency I / II. BARF1 is expressed in some cells in which lytic infection is reactivated. Such incorrect statements are everywhere.

ANSWER. We completely agree with this observation. In fact, BARF1 is an early lytic gene which is activated during the switch latent/lytic. However, the expression of BARF1 is consistently found in nasopharyngeal carcinomas and EBV-associated gastric carcinomas, in which EBV establishes latency I/II program (see references in the manuscript). It was previously reported that some lytic genes are expressed in cancer, establishing a type of interaction named “abortive lytic infection” in which a partial set of lytic products are expressed, among them, BARF1. In NPC, BARF1 commonly functions in parallel with LMP1, but in EBVaGC BARF1 is expressed in the absence of the latent protein, representing the main EBV oncogene in these tumors. Taken together, we speculate that BARF1 could acts as an oncogene in others epithelial cancers such as cervical carcinomas, although more studies are strongly warranted. According to our knowledge, only one study addressed BARF1 expression in cervical carcinomas (Algerian women). Anyway, some sentences were changed to clarify this point.

REVIEWER 1. To improve Table 1, the font size must be increased. There are many unnecessary entries in this table. For example, although the reference is described in Table 1, the space can be omitted by writing the reference number. Conclusion in the table should be explained in the text if it is really necessary. Also, part of the table has disappeared. Tissue can be Frozen or FFPE only. By making such an arrangement, it is possible to eliminate unnecessary information and increase the size of the font.

ANSWER. Many thanks for these observations. Table 1 was improved, and the font size was increased. References format was changed for numbers. Conclusion and Tissue type columns were eliminated.

Round 3

Reviewer 1 Report

All of figures were improved.

This manuscript is a resubmission of an earlier submission. The following is a list of the peer review reports and author responses from that submission.

Round 1

Reviewer 1 Report

COMMENTS FOR AUTHORS

The manuscript by Blanco et al. reviews available evidences regarding the role of EBV/HPV coinfection in cervical cancer development. I really appreciate the presented paper. Despite the achievements in term of primary and secondary prevention, cervical cancer remains a very big challenge given the very difficult penetration of screening and preventive programmes penetration in low income population. Furthermore, the biology of cervical cancer remains not completely understood with HPV playing a major, but not unique role in cancer development. In this context, EBV infection, which frequently carries on immune system impairment, is progressively emerging as a potential cofactor in cervical carcinogenesis, but only an attracting meta-analysis of available literature has been published until now. Therefore, the scientific relevance of the presented paper appears certainly high. On the other hand, some concerns should be raised:

  1. Native speaker revision seems mandatory, some specific paragraphs need to be carefully read to get the author message.
  2. Table 1 is required, and appropriate, but it must be rewritten adding a column named conclusions, briefly describing the main finding of each study.
  3. The review should follow PRISMA guidelines. The methods should be described specifying the search strategy; furthermore, a CONSORT diagram is required clarifying study selection.
  4. One of the main concerns of existing literature regarding HPV/EBV coinfection is represented by the not homogenous, and frequently not stated definition of EBV infection. The presence of high titres serum IgM against EBV may implies a recent persistent infection, and acute infection, etc.. However, the real viral activity is certainly the circulating viral genome load. Therefore, it should be useful in Table 1 to specify which studies investigated the circulating viral load, and clearly stating the limitation in the discussion section.   
  5. What about HIV coinfection? It is well known that active HIV infection provides the most significant immune disruption, and this infection is often associated with EBV, particularly in drug addicted, low incomes subjects. Therefore, all studies aimed at analysing the role of HPV viral coinfections in driving cervical carcinogenesis must include data regarding HIV status. This is another limitation of existing literature which does not help to definitely clarify the impact of simultaneous EBV/HPV infection. I suggest discussing this point, inviting future research to consider this bias in data analysis.

Reviewer 2 Report

This manuscript by Blanco and others is a review article on coninfection of EBV and HPV in cervical cancer or pre-cancer. It is well organized on the whole and therefore this can be accepted for publication after minor revisions as follows.

Page 2 line 68: Please mention that, in latency IIb, EBV expresses very limited amount of LMPs.

Page 8 line 255: I am afraid but I could not find Figure 3.